# A Simple Elimination of the Thermal Convection Effect in NMR Diffusiometry Experiments

**DOI:** 10.3390/molecules27196399

**Published:** 2022-09-27

**Authors:** Dávid Nyul, Levente Novák, Mónika Kéri, István Bányai

**Affiliations:** Department of Physical Chemistry, University of Debrecen, 4032 Debrecen, Hungary

**Keywords:** NMR diffusiometry, thermal convection, PGSE

## Abstract

Thermal convection is always present when the temperature of an NMR experiment is different from the ambient one. Most often, it falsifies the value of the diffusion coefficient determined by NMR diffusiometry using a PGSE NMR experiment. In spite of common belief, it acts not only at higher temperatures but also at temperatures lower than in the laboratory. Sodium alkyl-sulfate monomers and micelles in D_2_O solvent were used as model molecules measured at T = 319 K in order to show that thermal convection sometimes remains hidden in experiments. In this paper, we demonstrate that the increase in apparent diffusion coefficient with increasing diffusion time is a definite indicator of thermal convection. Extrapolation to zero diffusion time can also be used to obtain the real diffusion coefficient, likewise applying the less sensitive pulse sequences designed for flow compensation or the expensive hardware, e.g., sapphire or Shigemi NMR tubes, to decrease the temperature gradient. Further, we show experiments illustrating the effect of a long diffusion time in which the periodic changes of the echo intensity with gradient strength appear as predicted by theories.

## 1. Introduction

Temperature-dependent liquid NMR experiments always suffer from thermal convection that falsifies the physico-chemical parameters determined by means of different NMR methods to a smaller or larger extent. This phenomenon exceptionally affects NMR diffusiometry experiments performed at temperatures different from laboratory temperature. Textbooks, in general, deal with it or even describe the theory of the effects of coherent flow in general; however, they rarely warn everyday users to pay attention to this phenomenon [1,2]. Thermal convection, generated by the temperature gradient, is only a special case of coherent flows and differs from them in flow profiles and time-dependent behavior [3]. In this paper, we show a typical, easily noticeable appearance of thermal convection in diffusion experiments made at 46 °C. We emphasize the importance of measuring the observed diffusion coefficient as a function of diffusion time (usually referred to as *Δ*) and the careful analysis of the stimulated echo decay. As a consequence, a simple solution is tested experimentally to eliminate the effect of thermal convection without using complicated pulse sequences and expensive, special NMR hardware elements.

Thermal convection, first observed by Bénard and described by Rayleigh, usually arises when the bottom of an NMR tube is warmer than its top because the gas flow heats the tube from the bottom [4]. It starts when the temperature gradient reaches a certain value characterized by the critical Rayleigh number, *R*_c_ [5,6]. This number is determined by such geometrical parameters as the height of the liquid level (the lower, the better), the radius of the NMR tube (the narrower, the better), material constants as the coefficient of the kinematic viscosity (the larger, the better), the thermal expansion of the liquid and the thermal diffusion [4,5,6]. The estimated *R*_c_ numbers for a 5 mm standard glass NMR tube are 67.4 and 215.8 in the case of insulating and conducting walls, respectively [3]. Of course, these numbers are approximate and still affected by some other factors. According to Morris et al. in chloroform, only 0.3 K cm^−1^ is enough to cause convection, while for water, about 6 K cm^−1^ is an approximate limit; therefore, in experiments in aqueous solutions, it is rarely taken into account [6].

Hahn, in his classic paper entitled Spin Echoes, mentioned the effect of thermal convection and the self-diffusion on the decay of spin echoes in the presence of the permanent gradient of the static magnetic field (**B**_0_) [7]. Carr and Purcell presented not only the effect of self-diffusion on transverse relaxation experiments but warned of the effect of convection and showed the difference in intensities between the odd-numbered and the even-numbered echoes [8]. The advent of PGSE (Pulse Field Gradient Spin-Echo) NMR introduction of the time-dependent field gradient brought new dynamism into NMR diffusion experiments [9]. In the case of self-diffusion, the so-called Stejskal–Tanner equation is used to evaluate the apparent diffusion coefficient, *D*_app_ [9].
(1)$$\frac{I}{I_{0}}=\exp(-\gamma^{2}g^{2}\delta^{2}D_{\mathrm{app}}\left(\Delta -\frac{\delta }{3}\right))$$
where *γ* is the gyromagnetic ratio, *δ* is the length of gradient pulses, *Δ* is the diffusion time and *g* is the gradient strength. Stilbs’ review article comprehensively summarized the theoretical and practical aspects of NMR diffusiometry, including the stimulated echo method and detailed the historical way of application of NMR for studying translational motions [10]. It is very important to note that both the diffusion and the convection were mentioned equally as being the source of intensity decay of spin echoes beyond the natural relaxation. The appearance of commercially available gradient probe-heads dramatically increased the number of NMR diffusiometry experiments because this method became an easy and fast tool to obtain a hydrodynamic size of molecules, micelles and nanoparticles through the Einstein–Stokes equation [11,12,13]. Excellent review articles attracted chemists and biochemists from different fields to use NMR diffusiometry to determine the size of macromolecules and to clarify the interaction between small molecules and colloid particles. These review articles do not mention [14] or just shortly mention the effect of thermal convection, most probably because the experiments were supposed to be performed at the laboratory temperature [15,16,17].

There are basically two solutions for minimizing the effects of convection when we use temperature-regulated gas flow measuring diffusion at temperatures different from the laboratory one. The first is the reduction in the temperature gradient itself. The application of slow spinning of the sample can use the Coriolis effect [18] and centrifugal force to reduce the horizontal temperature gradient [6]. However, it introduces mechanical instability. Increasing the flow of cooling gas until the sample does not start to vibrate and using a narrower sample tube can also help [19,20]. Decreasing the sample length is also effective, although it may cause shim problems [19,21]. Specific design of the sample holder, e.g., a 5 mm NMR tube concentrically in a 10 mm one with high heat capacity liquid between them or simply loading the tube with glass wool may also be applied [6,22]. Using a sapphire tube reduces the coherent flow remarkably [5,6,21,23].

The other way is to refocus the effects of coherent flow along the gradient by specific pulse sequences [3,24,25,26,27,28,29]. These pulse sequences are effective; however, their application needs more NMR knowledge than common users usually have. A comprehensive summary of the convection effect on diffusion NMR from a practical point of view appeared in 2015, including the quantification of the problem, listing different strategies to avoid it and even determination of the rate of convective flow [6]. In that paper, the effect of a transverse temperature gradient is emphasized in detail as being a source of *z*-directed convection too [6,19]. As a consequence, it was shown that experiments maintained at temperatures lower than the laboratory also affected the echo decay by thermal convection in spite of widely accepted beliefs [6].

Furó et al. explicitly mentioned in their paper where NMR temperature imaging was described that the obvious indication of the convection being present was the diffusion time dependence of the determined apparent diffusion coefficient [5]. In another paper of theirs, beyond the essential description of thermal convection, they showed experimental data that diffusion coefficients increased linearly with increasing diffusion time according to Equation (2) [22].
(2)$$D_{\mathrm{app}}\approx D+v^{2}\Delta$$
where *D* is the real diffusion coefficient, and *v* is the rate of thermal convection. They even argued that the absence of diffusion time dependence is clear evidence for the absence of coherent flow (such as convection) disturbances [5]. In spite of this information, available mostly in specific NMR literature, only a few papers deal with thermal convection written in non-NMR journals. Studying the change of non-ionic micelle shape and size as a function of temperature by PGSE NMR, Furó et al. raised the possibility of using the diffusion coefficient extrapolated to *Δ* = 0 according to Equation (2). However, they rather preferred to apply a convection-compensated pulse sequence, because, in the presence of convection, the original Stejskal–Tanner equation is not valid [22]. The echo intensity in the presence of idealized convection (when *v* is a single parameter and its spatial distribution is not taken into account) can be given as (Equation (3)):(3)$$I∼\cos(\gamma \delta gv\Delta )\exp(-\gamma^{2}\delta^{2}g^{2}D(\Delta -\frac{\delta }{3}))$$
where the variables are the same as in Equations (1) and (2) [6,21,22,30]. The reason that Equations (1) and (3) are sometimes indistinguishable in a simple PGSE experiment is that the cos function of *g* (first term in Equation (3)) changes similarly to the Gaussian function of *g* (second term in (3)) when *v*^2^*Δ* << *D* and *δ* << *Δ*. Model calculations in the paper by Morris et al. show that at higher diffusion time, the uncertainty of *D*_app_ obtained by fitting the echo decay by Equation (1) increases. In the case of an extreme long *Δ* negative echoes may appear in PGSE experiments [6].

In this paper, we show how these mostly theoretical considerations show up or hide away in a series of experiments. In an industrial project, we planned extensive PGSE NMR measurements in order to determine the dependence of the size and shape of ionic micelles on the length of the hydrophobic chain (from 8 to 16). We studied the diffusion of the sodium alkyl-sulfates at 319 K, well above the laboratory temperature, in order to avoid solubility problems. The increase in *D*_app_ with increasing *Δ* was experienced in almost all cases. By intentionally applying higher diffusion times than the optimal ones we could even experimentally demonstrate a negative and periodic change in intensity of the stimulated echo vs. *g*^2^ curves, as the above-mentioned considerations and model calculations show [6,22]. We also confirm that the extrapolation of *D*_app_ to *Δ* = 0 gives real values of diffusion coefficients. We show that by applying the standard PGSE pulse sequences without convection compensation, the problem can be recognized and avoided by measuring the diffusion time dependence.

## 2. Results and Discussion

Figure 1 shows the typical ^1^H NMR spectra of sodium decyl-sulfate (NaDS) as a function of concentration.

There is practically no difference between the spectrum of DS monomers and micelles formed from the monomers. It is surprising that the hydrated monomers and non-hydrated micelles do not differ in chemical shifts of the protons in CH_2_ groups. From this, we can conclude that the chemical shifts, in this case, are not good parameters to check the presence of micelles in the solutions in spite of our expectations.

Since micelles are larger than the monomers, the apparent diffusion coefficient of the surfactants in the solution is different below and above the *cmc*. DOSY experiments provide a precise way to determine the diffusion coefficient of these components and, therefore, the value of *cmc* [10,31]. In Figure 2, we show the apparent self-diffusion coefficient of NaDS measured in D_2_O solutions of different concentrations at 319 K. The measured self-diffusion coefficients show significant but different dependences on the diffusion time.

The intercepts of the curves show the expected tendency with the concentration. Until 0.029 mol kg^−1^ concentration, the *D*_app_ values extrapolated to *Δ* = 0 are constant. At concentrations higher than 0.029 mol kg^−1^, the extrapolated diffusion coefficients start to decrease, probably because of the appearance of micelles in an increasing concentration ratio. Figure 3 shows the dependence of the intercept on the concentration. The estimated critical micelle formation concentration (*cmc*) is approximately 0.035 mol kg^−1^, which is in good agreement with the literature data [32].

The slopes of the lines in Figure 2. do not show as clear a dependence as the intercepts do. There is essentially an increasing trend, except for one concentration. In the most diluted solution, *D*_app_ slightly decreases with increasing diffusion time, which indicates a kind of hindered diffusion. It may be explained that observing the motion of the molecule for a longer time makes the collisions between them more probable than in a shorter time. Therefore, the random walk conditions are not fulfilled [14,33]. The increasing trend of *D*_app_ with *Δ* at larger concentrations is the probable indication of thermal convection, although the exchange process between the monomers and micelles may also be considered [16,34]. If the reason is thermal convection, then the slope is in connection with the convection velocity (*v*) according to Equation (2). Further analysis of the slope is out of the scope of this paper because there are very detailed considerations on it in the literature and where the convection rate is the parameter needed [6,22].

Since above the *cmc*, both monomers and micelles are present, with a ratio depending on the total concentration, structural information for micelles from the obtained diffusion data is not straightforward. Consequently, the obvious evidence that with extrapolation, the real diffusion coefficient is given is very important since the most important parameters, the size and shape, of micelles are to be calculated from the convection-free diffusion coefficients. In order to exclude the effect of monomer-micelle dynamic equilibrium, we made experiments with sodium octyl-sulfate (NaOS) and sodium hexadecyl-sulfate (NaHDS) in the same molar concentrations at about 0.07 mol kg^−1^. The *cmc* values for NaOS, NaDS and NaHDS are ~0.129 (extrapolated), 0.033 and 0.0006 mol kg^−1^, respectively [32]. It means that in the case of NaOS, practically only monomers, while in that of NaHDS, only micelles are present. The results are shown in Figure 4. The apparent diffusion coefficients of octyl- and decyl-sulfates show a conspicuous linear increase with the diffusion time, but for the hexadecyl-sulfate and the water, this dependence is negligible. These experimental results lead to the conclusion that the phenomenon causing the linear increase in *D*_app_ as a function of diffusion time tends to be not of chemical origin but rather physical.

The explanation is that the influence of thermal convection is larger for slowly-diffusing molecules [5,6,22]. The effect of convection is almost negligible for small molecules such as HOD, larger for NaOS monomers and even larger for micelle-containing NaDS solution. In the case of NaHDS, the high viscosity caused by the big micelles removes the convection effect [6].

In order to confirm the presence of thermal convection further and the reality of *D* extrapolated to *Δ* =0, we tested the behavior of the 0.07 mol kg^−1^ solution of NaDS in a sapphire NMR tube (the thermal conductivity is 25 times that of borosilicate) and in a normal NMR tube but implementing a pulse sequence, which reduces the effect of thermal convection developed by Jerschow et al. [24,25]. The reduction of the effect of thermal convection is clearly seen in Figure 5.

As expected, using both a sapphire NMR tube and in a normal 5 mm NMR tube, in the experiments applying a convection-compensated pulse sequence, practically no diffusion time dependence was observed [5,6,15,22,25]. Furthermore, it is visible that the intercept of the three lines is the same. Therefore, the linear extrapolation of *D*_app_ values to zero diffusion time results in the real diffusion coefficient.

Equation (3) refers to echo decay under the effect of convection; it contains a cos(*γδgvΔ*) function, which may result in periodic changes in the primary experimental echo intensity vs. *g*^2^ function. This effect of the cos term at a short diffusion time is not visible; the *I*/*I*_0_ vs. *g*^2^ function can be well fitted with Equation (1) in accordance with our experiences [6,22]. In Figure 6A, the results of the least square fit of Equation (1) on the echo decay data are seen at short diffusion times. One can see very good fit up to 50 ms (in this case *δ* = 4 ms). Only at large gradient values (see insert) do small deviations appear between the measured and calculated values. Applying a 120 ms diffusion time instead of the usual maximum of 60 ms resulted in us detecting the experimental periodic changes in peak intensities (Figure 6B). The amplitude is not large, but it is visible. Equation (3) can reproduce this periodicity (red line), but the fit is not very good, pointing out the rough approximations applied for the deduction of this equation [6,22,35]. The fits with Equation (1) are spectacularly worse than that with Equation (3) in this case (blue line). Further, we can observe (in Figure 6B) that at the lower gradient values, both equations fit well. It illustrates that the independent determination of the convection rate (*v*) and the diffusion coefficient (*D*_app_) is not possible only by parameter fitting procedures.

## 3. Materials and Methods

The sodium alkyl-sulfates were prepared at the Eötvös Loránd University Budapest and characterized by ^1^H NMR spectroscopy (Figure 1) [32]. The integrated intensity of the C-H protons was used to check the purity of the surfactants. Every measured sample was freshly prepared and checked before the diffusion experiment in order to avoid the effect of partial hydrolysis. Sodium-decyl-sulfate was mostly used. However, to verify some statements, we used the results obtained on other surfactants indicated above.

Approximately 0.07 mol kg^−1^ solutions of three NaOSO_2_(CH_2_)_n_CH_3_ were prepared by weight (n are 9 NaDS, 7 NaOS and 15 NaHDS) and measured in deuterium oxide solution at 319 and at 298 K (laboratory temperature). A total of 500 uL solution was used as a standard volume in each case to reach good shim. Norell © type 5 mm NMR tubes were applied in all cases, but in one series of experiments, a sapphire tube was used.

A Bruker Avance II 400 MHz NMR spectrometer equipped with a 5 mm ^1^H–X inverse gradient probe head was used. The temperature was regulated by a Bruker BCU 4 cooler using dry air flow with a rate of 800 L/h. Under the standard Bruker TopSpin 2.1 software, a 2D pulse sequence (ledbpgp2s provided with the spectrometer) was used without modification for diffusion measurement. Stimulated echoes were recorded by applying LED (low eddy current delay) and bipolar gradient pulses, and two spoil gradients [36]. Usually, from 5% to 95% of the available gradient strength is used with 64 steps. The distances between consecutive gradient pulses were applied in the square mode. The gradient value was calibrated for D_2_O as *D* = 1.9 × 10^−9^ m^2^ s^−1^ (298 K) [37]. In each experiment, *δ* and *Δ* in Equation (1) were kept constant, and *g* varied. The apparent diffusion coefficients were evaluated from the decay of individual peaks separately by using MestreNova 8.1© software, fitting Equation (1) to the experimental data. The average values of these apparent *D* values were used for further analysis. The decrease in stimulated echoes measured by using an extra-long diffusion time (*Δ*) was fitted by Equation (3) as well. In one series of experiments, we adapted a modified pulse sequence, named double-stimulated-echo pulse sequence, published by Jerschow and Müller dedicated to suppress “convection artifacts” in the PGSE experiments [24]. The real diffusion coefficients were obtained by linear regression from *D*_app_ vs. (*Δ* − *δ*/3) curves, as illustrated in the Appendix A.

## 4. Conclusions

Spectroscopic and spectrometric techniques, including NMR, have been and will be the superior tools for structure determination of species from small molecules to colloid-size particles. NMR diffusiometry is a source of many valuable physico-chemical parameters of molecules and association interactions between large and small molecules. In most of the cases, these data originate from the determination of the diffusion coefficient using optimal diffusion time PGSE experiments. However, diffusion NMR experiments suffer from thermal convection when the measuring temperature is different from the room temperature. There are successfully applied NMR hardware solutions that are rather expensive and software solutions that are less sensitive and complicated for compensating for the effect of thermal convection. In this paper, we showed, with extensive experimental work, that by applying the usual hardware and software tools, the analysis of diffusion time, the dependence of apparent diffusion coefficients is equivalently suitable to obtain real diffusion coefficients. The apparent diffusion coefficients can be determined in the presence of convection by fitting the Stejskal–Tanner equation (1) on the experimental echo decay data within the limit of certain NMR parameters. It is important that the good least square fits do not indicate the absence of thermal convection in optimal diffusion times. The thermal convection only becomes visible from experiments by using extremely long diffusion times. The extrapolation of the obtained apparent diffusion coefficient to zero diffusion time for the real diffusion coefficient can be determined and used for obtaining structural information. Although it is believed that using D_2_O avoids thermal convection, we showed that it is not evident. Measuring the diffusion time dependence of the observed diffusion coefficients is always required.

## Figures and Tables

**Figure 1 molecules-27-06399-f001:**
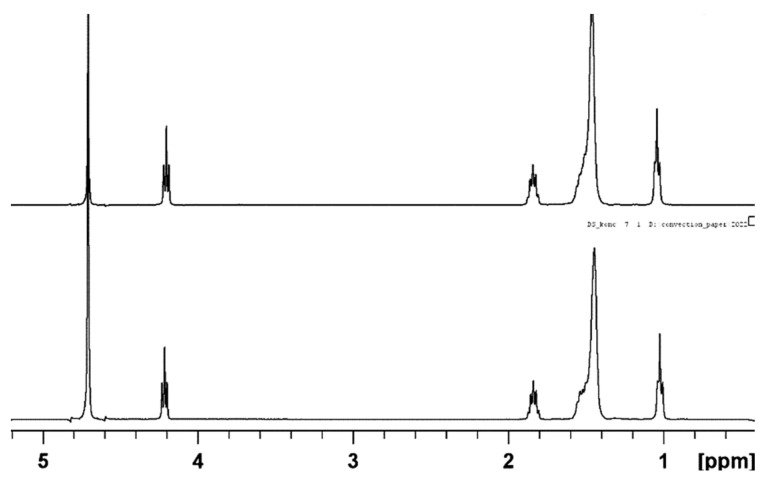
^1^H NMR spectra of sodium decyl-sulfate (NaDS) below *cmc* (5 × 10^−3^ mol kg^−1^, upper spectrum) and above *cmc* (5.6 × 10^−2^ mol kg^−1^, lower spectrum). Peaks assignment: 4.71 ppm HDO, 4.22 ppm CH_2_ (1), 1.84 ppm CH_2_ (2), 1.4–1.6 ppm CH_2_ (3–9) and 1.03 ppm CH_3_.

**Figure 2 molecules-27-06399-f002:**
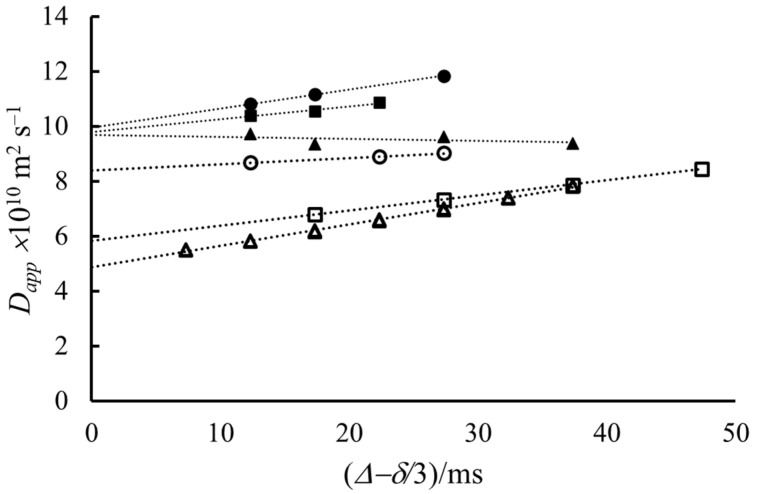
The apparent self-diffusion coefficients measured at different concentrations as a function of the diffusion time (*Δ*) at 319 K. Concentrations: ▲ 0.005 mol kg^−1^, ● 0.015 mol kg^−1^, ■ 0.029 mol kg^−1^, ○ 0.047 mol kg^−1^, □ 0.059 mol kg^−1^, Δ 0.074 mol kg^−1^.

**Figure 3 molecules-27-06399-f003:**
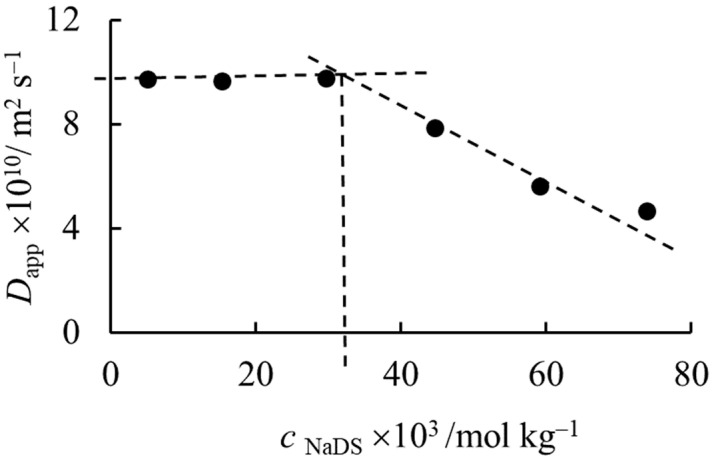
Dependence of the intercept (*D*_app_) of lines in Figure 2. on the concentration of NaDS. The intersection of the lines gives the *cmc*.

**Figure 4 molecules-27-06399-f004:**
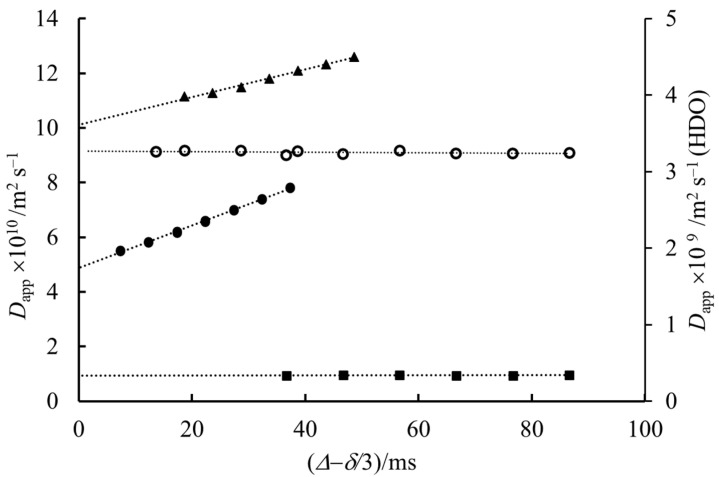
The dependence of the apparent diffusion coefficient of molecules present on the diffusion time at 319 K. ■ NaHDS ● NaDS and ▲ NaOS while ○ HDO.

**Figure 5 molecules-27-06399-f005:**
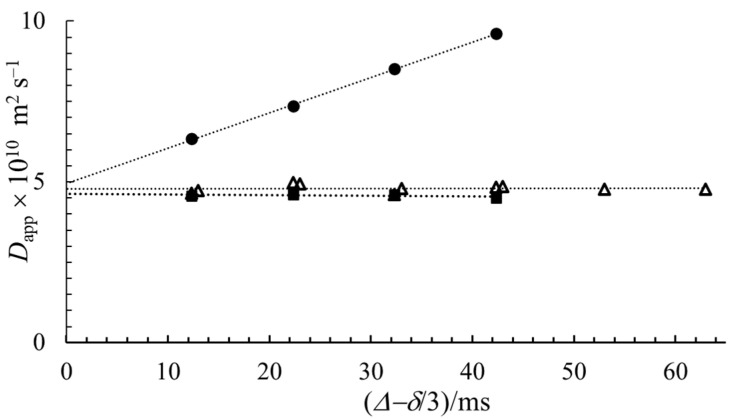
The apparent diffusion coefficient of NaDS as a function of diffusion time at 319 K. ● normal 5 mm NMR tube, ■ in a sapphire tube applying the standard pulse sequence without convection reduction and Δ with pulse sequence with convection compensation.

**Figure 6 molecules-27-06399-f006:**
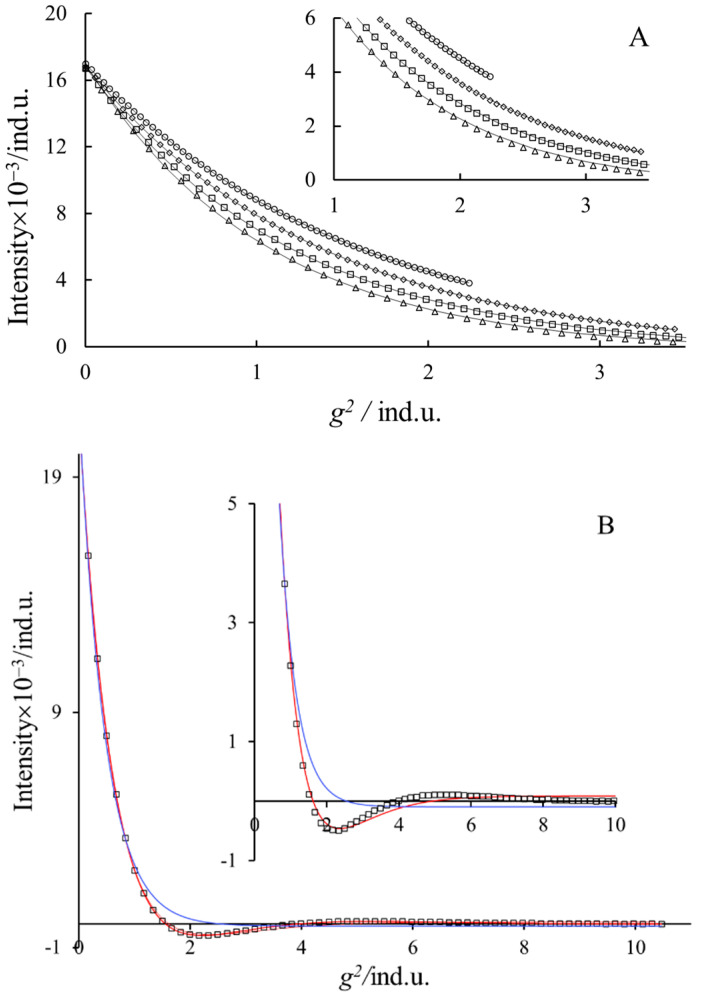
The decrease in the stimulated echo intensity with the increasing strength of the gradient square (0.07 mol kg^−1^ NaDS, T = 319 K, *δ* = 4 ms). The inserts are the enlargements of the wrapped part of the curves. (A) The best fit of Equation (1) *Δ* = [circle = 20, diamond = 30, square = 40 and triangle = 50 ms] (B) *Δ* = 120 ms. The blue and red lines are the best fits of Equations (1) and (3), respectively.

## Data Availability

Not applicable.

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
