# Peer review of "A Simple Elimination of the Thermal Convection Effect in NMR Diffusiometry Experiments"

_molecules, 2022, doi:10.3390/molecules27196399_

Round 1

Reviewer 1 Report

In this manuscript the authors deal with temperature gradient driven convective flow that make diffusion measurements inaccurate. In the manuscript is in the introduction given detailed overview of the relevant literature on this subject. Presented is the background of the effect and some solutions for its reduction. In the results section the authors show experiments that prove that dependence (increase) of apparent diffusion coefficient with diffusion time is an indicator of temperature driven convective flow and that extrapolation of the apparent diffusion coefficient to zero diffusion time yields true diffusion coefficient.

This is definitely a very interesting study. I recommend its publication after a minor revision that includes the answers to the following questions/comments

General comments

1.       To my understanding you are dealing here with two effects. First is a direct effect and in simply dependence of diffusion on temperature, while the second effect is indirect and is due to convection flow induced by temperature differences in the sample. This second effect is in a way analogous to perfusion in in vivo systems which causes intravoxel incoherent motion that can be detected as diffusion with low b-values. For this reason, the term “apparent diffusion coefficient is used” or ADC. There, a recipe to avoid from this effect is to use sufficiently high b-values that well suppress all signals originating from flowing liquids. I would assume that this recipe works also for convection induced flow. Can you comment on this? The second question is on the ratio between the direct and the indirect effect on diffusion measurement. Which of these two is higher and how experimental conditions influence on this ratio.

2.       In addition to convective flow that reduces echo signals there is also signal attenuation due to internal gradients (susceptibility effects). These are usually present in heterogenous systems where you also have temperature gradients that drive convective currents. How can you distinguish between these two effects? Can you control them? In general, both of them lead to higher echo signal attenuations and therefore overrated diffusion (D_app).

Minor comments

3.       Line 37, complicate -> complicated

4.       Line 94, consequences -> consequence

5.       Line 143, From we … -> From this we …

6.       Line 147, … for determination of cmc. Explain what the term cmc! (is this concentration of micels?)

7.       Line 216, augment of cosine function is not only gradient g. To describe other factors, insert proportionality factor, e.g., alpha or alternatively write all factors that appear in Eq. 3

8.       Line 245, some certain -> some

Author Response

Q1 "There, a recipe to avoid from this effect is to use sufficiently high b-values that well suppress all signals originating from flowing liquids. I would assume that this recipe works also for convection induced flow. Can you comment on this?"

What we know about MRI (it is not too much) that b-value is defined as SS0exp(-bD). We experienced that at larger Δ-value the effect a thermal convection is larger. Since the Stejskal-constant is b=γ2 δ2 G2 (Δ-δ/3) it means that application of higher b-value results in higher contribution of thermal convection if we increase b using higher Δ time. We suppose that  since Δ is only one component of b and this is the explanation of our  observations. This may be the reason that our experiences differ from the Reviewer's suggestion.  Anyhow we will check the relevant literatures for our further work.

Q1(b) "The second question is on the ratio between the direct and the indirect effect on diffusion measurement. Which of these two is higher and how experimental conditions influence on this ratio."

A1(b) In our other work connected to this manuscript does not deal with the effect of temperature on diffusion coefficient. Only we elevated the temperature for solubility reasons. Furó et al. (Hedin, N.; Yu, T. Y.; Furó, I. Langmuir 2000, 16, (19), 7548-7550) suggest that for temperature dependence studies the convection reduced pulse sequence should rather be applied than the diffusion time dependence. We do not have our own experiences about the ratio of contributions. According to our knowledge the larger the diffusion time the larger the effect of thermal convection.

Q2 "How can you distinguish between these two effects? Can you control them? In general, both of them lead to higher echo signal attenuations and therefore overrated diffusion (D_app)."

A2 The question is about the internal gradient effect. In the present case we worked in "well shimmed" high field NMR with homogeneous samples. We did not have to deal with this in these experiments. However, we usually work with low-field relaxometers and with porous materials. In that case it is a serious problem where there is remaning gradient even for homogenous samples . Our solution at the moment is that we calibrate it by measuring a standard sample of known diffusion coefficent. There are solutions by means of fitting more complicated equations as described in Ardelean's and Kimmich's paper (  Ardelean, I.; Kimmich, R., Principles and Unconventional Aspects of NMR Diffusometry. In Annual Reports on NMR Spectroscopy, 2003; 49, 44-111). 

We corrected the manuscript accordin to suggestion in the comment 3.4.5.

Q6 "Explain what the term cmc!"

A6 cmc is an approximate parameter for characterization of micellar systems. It is the concentration under which the presence of micelles is negligible. In colloid science it is said that the micell conentration is zero under cmc. Of course, since it is a chemical equilibrium both monomers (Mo) and micelles (Mon) are always present. The explanation of the goodness of cmc is that in the equilibrium nMo = Mon the n-value is normally high (50-100) therefore the concentration range where the micelles start to appear is narrow. Above cmc there is another approximation: the monomer concentration is constant and the value is the cmc

We correcte the manuscript according to notes 7 and 8. 

General note:

Since translation diffusion happens during (Δ-δ/3) and we plotted our values in Figures  we changed title of the horizontal axes in Figures 2,4 and 5 accordingly. Normally δ/3 is negligible but we did not neglect it.

Reviewer 2 Report

NMR diffusion experiments are a powerful tool used to estimate the hydrodynamic size of different kind of molecules, conjugates and/or aggregates in solution, using the Einstein-Stokes equation. It has long been known that thermal convection is a factor that affect the diffusion measurements, in particular when the measurement temperature significantly differs from room temperature. Previous reports had demonstrated that the apparent diffusion coefficient increases linearly with the increasing of the diffusion time parameter of the NMR experiment (BIG DELTA). Even though it has previously been postulated that the real diffusion coefficient could be extrapolated at BIG DELTA = 0 ms from plots of measured diffusion coefficient vs BIG DELTA, this strategy has not been fully addressed before experimentally. The authors demonstrated the utility of this approach using sodium decylsulfate as test substance in D2O and at 46 C. The results of using this approach were similar to those obtained using other alternative approaches (such as expensive sapphire NMR tubes and non-standard NMR pulse sequences that require setup by experienced users).

The manuscript is well written and presented in a clear and well-structured manner. The introduction to the topic is clear, and the references are pertinent. The figures are adequate, but some minor recommendations are given below to improve clarity. The methodology and the results are clearly described. The statements and conclusion are drawn coherent and supported by the data and listed citations. I consider this work addresses an interesting strategy to overcome a well know problem in the measurement of diffusion coefficients by NMR spectroscopy, and may be of interest to the readers of this journal; thus, I suggest accepting it for publication with some comments (see below).

Comments:

In some of the plots of Figure 2 only three data points where used. Could the authors please explain this selection of data, and supply the fitting parameters of all the diffusion measurements as supporting information. 

Could the authors please explain the large magnitude differences in the x- and y-axes between the plots of panel A and B? 

Could the authors please indicate the value of the gradient they calibrated (mentioned in line 259-260)

Figure 2: note that the x-axis ticks are missing

Figure 4: add units in the right side y-axis

Figure 6: note that the x-axis ticks are difficult to observe since they are overlapping with the curves. The x-axis could be moved further down to intercept the y-axis at -1000.

Figure 6: note that the A and B panels are not labelled (as indicated in the figure legend)

Figure 6 legend: please use different markers in the plots of panel A to differentiate the data obtained with different BIG DELTA parameters. Add the corresponding information in the figure legend. 

Figure 6 legend: note that the BIG DELTA and small delta symbols are missing 

Some subscripts or superscripts have to be fixed in the text: “D2O” (see line 12 & 148), “kg-1” (see lines 139, 140, 158, 161 & 235), “1H” (see line 139 & 314), etc.

Concentrations are given in “mol kg-1” whereas CMC values are given in “mmol kg-1”(page 5 lines 161 and 186). It would be recommended to use the same units in both cases.

Please note that the term “sulfate” is recommended by IUPAC instead of “sulphate” (it shows up 6 times in the document)

In page 5, line 157: where it reads “0.029 kg mol-1” it should read “0.029 mol kg-1”

In page 7, line 216: where it reads “ubder” it should read “under”

In page 8, line 248: where it reads “NsOS” it should read “NaOS”

Author Response

Note 1

"In some of the plots of Figure 2 only three data points where used. Could the authors please explain this selection of data, and supply the fitting parameters of all the diffusion measurements as supporting information. "

We planned the experiments on Figure 2 for simply to check the critical micelle formation concentration (cmc) before starting our projects for the size determination. We routinely measure always minimum at three diffusion time. During this routine work we observed the increasing trends. Then we started a more careful investigation at higher concentrations. Actually the cmc is known for this NaDS which is not an exact thermodynamic parameter only a (narrow) range of concentration where the micelle formation starts. This the reason that we did not measure more points under this value where the monomers are dominant.

We made a Supporting Information but for saving times we have chosen selected diffusion curves to illustrate the statistical parameters of fitting because the MestreNova, we used for fitting, does not provide details, only one number is given. We have chosen randomly not from the best situations. Hope, these data fulfill the requirements. 

Note 2

"Could the authors please explain the large magnitude differences in the x- and y-axes between the plots of panel A and B?"

Yes. In one the B we used order magnitude in Excel in the A not. We corrected it now. 

Note 3

"Could the authors please indicate the value of the gradient they calibrated (mentioned in line 259-260)" 

No, we cannot because it is not specified exactly for the probe head. The procedure used is as follows. Practically, before a series of DOSY experiment we  measure the diffusion coefficient of D2O (99,9 %). Then we use the MestreNova post processing software, where there is a constant k to be modified until we obtained the literature value of D. Then we use this correction factor for the rest of experiments. We tested this method even for hetero nuclei (23Na, 7Li, 31P) and it gives the correct value of D for the standards. (I think it is approximately 50 Gauss/cm)

Note 4,5

The axes are corrected.

Note 6

We moved the ticks to outside of the axis. It is better visible now. We did not want to move the horizontal axis down because in this form the negative echo intensity is more conspicuous. When we observed it first  we suspected it was mistake.

General note:

Since translation diffusion happens during (Δ-δ/3) and we plotted our values in Figures  we changed title of the horizontal axes in Figures 2,4 and 5 accordingly. Normally δ/3 is negligible but we did not neglect it.

Reviewer 3 Report

To the Editor of Molecules

Review of the Ms. Ref. No.: molecules-1924049

The manuscript entitled „A Simple Elimination of the Thermal Convection Effect in NMR Diffusiometry Experiments” by Dávid Nyul , Levente Novák , Mónika Kéri, and István Bányai describes the effect of thermal convection on self-diffusion coefficient determined by NMR diffusometry methods. The authors describe how the thermal gradients in the sample affect the determined by simple PGES NMR experiment diffusion coefficients of investigated systems. Based on literature data and approaches provided and proposed by others, for example, Furo et al., the authors show how the reader can deal with thermal artifacts in their own experiments. The paper can be treated like a good manual for those who work with diffusion NMR or needs to measure the self-diffusion coefficients in a number of different systems as a function of temperature. The article has a good basic didactic layout and can be usefull for many readers. As such, I can recommend it for publication after minor revision. The authors should mark A and B in figure 6 as they use this notation in the text. The font size of the axis description in figure 1 should be increased, and the particular lines should be assigned for better didactic purposes. I would also suggest adding one figure illustrating the process of extrapolation of the apparent diffusion coefficient to diffusion time equal 0, as this is described in the text, and the main idea of dealing with thermal convection artifacts is based on that approach.

Author Response

"The authors should mark A and B in figure 6 as they use this notation in the text. The font size of the axis description in figure 1 should be increased, and the particular lines should be assigned for better didactic purposes. I would also suggest adding one figure illustrating the process of extrapolation of the apparent diffusion coefficient to diffusion time equal 0, as this is described in the text, and the main idea of dealing with thermal convection artifacts is based on that approach."

We changed Fifure 1. accordingly and wrote the assignment in the caption of Figure. We made a Supplementary Information which shows the lienar regresiion on the apparent D vs. diffusion time plots.

General note:

Since translation diffusion happens during (Δ-δ/3) and we plotted our values in Figures  we changed title of the horizontal axes in Figures 2,4 and 5 accordingly. Normally δ/3 is negligible but we did not neglect it.